# Host plant phylogeny predicts arbuscular mycorrhizal fungal communities, but plant life history and fungal genetic change predict feedback

Robert J. Ramos[1,2]*, Brianna L. Richards[3], Peggy A. Schultz[4,5], James D. Bever[4,6]*

**1** Cooperative Institute for Research in Environmental Sciences (CIRES), University of Colorado, Boulder, Colorado, United States of America, **2** The Environmental Data Science Innovation & Impact Lab (ESIIL), University of Colorado, Boulder, Colorado, United States of America, **3** National Ecological Observatory Network, Battelle, Boulder, Colorado, United States of America, **4** Kansas Biological Survey and Center for Ecological Research, The University of Kansas, Lawrence, Kansas, United States of America, **5** Environmental Studies, The University of Kansas, Lawrence, Kansas, United States of America, **6** Ecology and Evolutionary Biology, The University of Kansas, Lawrence, Kansas, United States of America

\* robert.ramos@colorado.edu (RJR); jbever@ku.edu (JDB)

## Abstract

Symbioses exert strong influence on host phenotypes; however, benefits from symbionts can increase or degrade over time. Understanding the context-dependence of reinforcing or degrading dynamics is pivotal to predicting stability of symbiotic benefits. Host phylogenetic relationships and host life history traits are two candidate axes that have been proposed to structure symbioses. However, the relative influence of host evolutionary history and life history on symbiont composition, and whether changes in symbiont composition translate into stronger mutualistic benefits is unknown. We tested the influence of plant phylogenetic relationships and plant life history on the composition of arbuscular mycorrhizal (AM) fungi, perhaps the most ancestral and influential of plant symbionts, and then tested whether AM fungal differentiation resulted in improved mutualism as expected from coadaptation. We constructed mycobiomes composed of seven AM fungal isolates derived from tallgrass prairie and grew them for two growing seasons with 38 grassland plant species. We found that host phylogenetic structure was a significant predictor of the composition of AM fungal communities and the genetic composition of AM fungal species, patterns consistent with phylosymbiosis. However, the phylogenetic structure of AM fungi failed to translate to improved benefits to their host. While AM fungi generally improved plant growth and mycorrhizal feedback was generally positive, the strength of feedback was not predicted by plant phylogenetic distance. The composition of the AM fungal community and genetic composition within AM fungal species were also significantly influenced by plant life history and feedbacks between early and late successional species were generally positive. Interestingly, positive mycorrhizal

**Data availability statement:** There are two repositories for associated data. All code and data tables are provided in the Open Science Framework (OSF) project osf.io/naxmt/: "Limited inference from phylosymbiosis ". The raw sequencing files for the experiment are stored in the NCBI Sequence Read Archive (SRA) under the BioProject PRJNA1200747: "Ecological Dynamics of the Plant-Arbuscular Mycorrhizal Fungal Mutualism".

**Funding:** This work was supported by the following grants: DEB-1556664; RJR, JDB, PAS, National Science Foundation; https://www.nsf.gov/; OIA-1656006; RJR, BLR, JDB, PAS; National Science Foundation; https://www.nsf.gov/; DBI-2027458; JDB, PAS, National Science Foundation; https://www.nsf.gov/; BII-2120153; JDB, PAS, National Science Foundation; https://www.nsf.gov/; DBI-2153040; RJR, National Science Foundation; https://www.nsf.gov/. The sponsors or funders did not play any role in the study design, data collection and analysis, decision to publish, or preparation of the manuscript.

**Competing interests:** The authors have declared that no competing interests exist.

**Abbreviations:** AM, arbuscular mycorrhizal; ASVs, amplicon sequence variants; LRR, log mycorrhizal responsiveness; PerMANOVA, Permutational Multivariate Analysis of Variance; PGLMM, phylogenetic generalized mixed model.

feedback was predicted by changes in genetic composition of the two most abundant AM fungal species, not by changes in species composition. Positive mycorrhizal feedbacks across life history can mediate plant species turnover during succession and suggests that consideration of mycorrhizal dynamics could improve ecosystem restoration.

## Introduction

Symbioses exert strong influence on host phenotypes. Symbionts are particularly important to plants, with 80% of all terrestrial plant species, including most crop plants, associating with arbuscular mycorrhizal (AM) fungi [1]. This symbiosis has been identified as ancestral to all root symbioses [2] with arbuscules predating the origin of roots [3,4]. AM fungi mediate host acquisition of soil resources, particularly phosphorus, and can have a large influence on plant community dynamics [5–9], illustrating the critical importance of symbionts to host biology.

Host phylogeny has been posited as an important factor structuring symbiont composition and function [10]. Phylogenetic structure of symbiont composition on symbiont species or genetic variants has been called phylosymbiosis. Phylosymbiosis has been frequently observed in gut microbiota of animals. However, there has been no comprehensive test of plant phylogenetic influence on AM fungal composition to date. Moreover, knowledge of the consequences of changes in composition on fitness of hosts is limited. It is tempting to assume that differentiation in symbiont composition results in improved symbiont functioning for their hosts (i.e., coadaptation), but fitness effects from host-specific differentiation of symbionts can result in positive or negative feedback [11] corresponding to strengthening or deteriorating host fitness with phylosymbiosis. While several studies have found evidence of reinforcing coadaptation of symbionts on their hosts [12,13], others have not [14]. The benefit of AM fungi to their hosts has been shown to decline [15,16] or increase [17,18] with host-specific differentiation of symbionts. Whether host phylogenetic relationships influence the likelihood of positive, reinforcing feedback remains an open question.

Host traits unrelated to phylogeny could also structure symbiont differentiation on hosts. While the candidate host traits driving symbiont differentiation are many, the major axis of plant trait variation has been shown to align with the fast-slow/acquisitive-conservative continuum of life history strategies [19,20]. Plant life history has also been shown to be important for plant–AM fungal interactions. Specifically, slow-growing late successional plant species have been shown to be more responsive to AM fungi [21,22], to be more sensitive to AM fungal identity [23], and to benefit particularly from AM fungi found in undisturbed sites dominated by late successional plant species [24,25], potentially generating positive feedback [18]. Such trait mediated AM fungal feedbacks could have important consequences for coexistence and turnover during succession.

Plants acquire AM fungal symbionts from the environment each generation—there is no vertical inheritance. While AM fungal species have been shown to be

functionally variable [8,26], genetic variants within AM fungal species can also vary in their ecology [27,28]. AM fungi are multinucleated, and different nuclei within AM fungal individuals can have substantially different genetic content [29–31]. Parent AM fungal cells can segregate nuclei, passing different nuclei to offspring, and these segregants have been shown to differ in their impacts on host plants [27,28,32–34]. The relative importance of changes in species composition versus genetic composition within species in driving functional consequences of AM fungal dynamics has not been tested.

To evaluate how plant host identity, phylogeny and life history influences AM fungal composition and resulting feedbacks, we work within the tallgrass prairie system in which we have characterized isolates of native AM fungi and their impact on hosts. Late successional prairie plant species have been found to be more responsive to AM fungal presence and more sensitive to AM fungal identity than early successional native or non-native plant species [21,23,35,36]. As a result, reintroduction of native AMF has been shown to improve prairie restoration quality [37,38]. How prairie plants affect AM fungal composition and the nature of the resulting feedback are less well understood.

To test how plant host identity, plant host phylogeny and plant life history shape the composition and function of symbiotic AM fungal communities, we grew a common community of AM fungi, a constructed synthetic mycobiome, with replicates of 38 plant species (Fig 1a) representing a phylogenetically diverse group of early and late successional host prairie plants (S1 Fig and S1 Table). All plants were grown in a common greenhouse environment to remove potential confounding environmental effects [10]. We monitored the changes in AM fungal composition over two growing seasons using large subunit of the ribosomal gene (rDNA) amplicon sequencing [40–42]. As rDNA of AM fungi are highly polymorphic

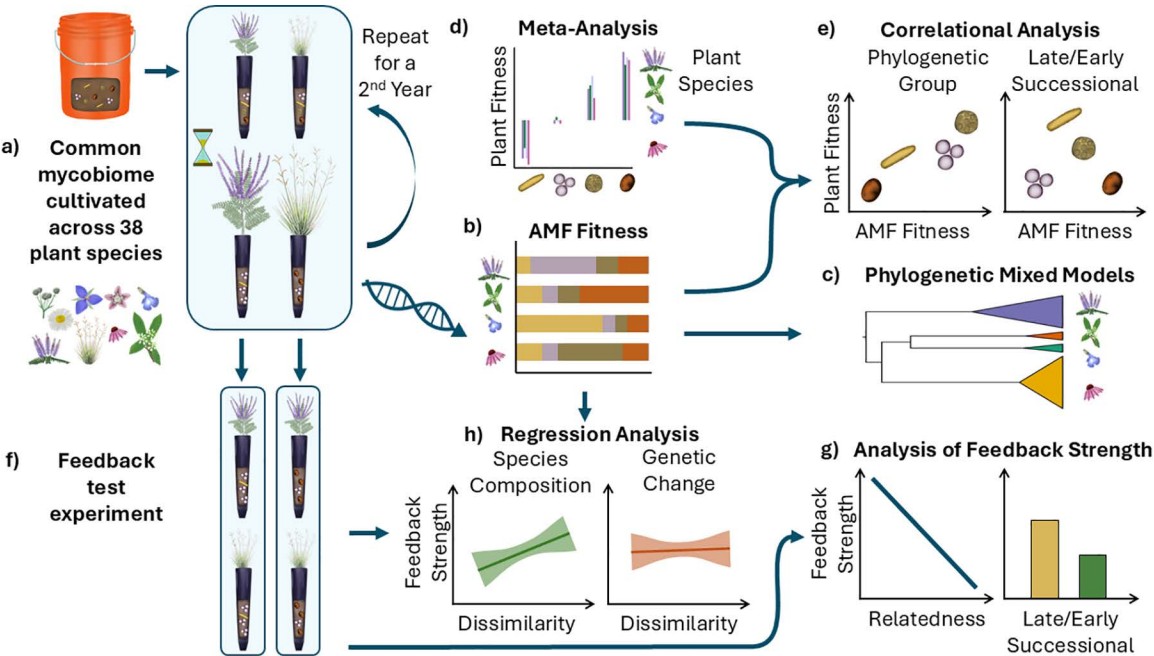

**Fig 1. Conceptual figure of the experimental design.** A common AM fungal community composed of mixtures of characterized isolates was grown in four replicate mesocosms with 38 plant species for two growing seasons **(a)**. Changes in relative abundance of AM fungi were monitored using amplicon sequencing **(b)** and used to test for plant phylogenetic and life history impacts on AM fungal species and genetic composition **(c)**. The effect of the AM fungal inoculum strains used in this experiment on plant species was characterized using meta-analyses of prior studies **(d)**. Correlations of AM fungal relative fitness and fungal impacts **(e)** on host phylogenetic and life history groupings were tested as one measure of mycorrhizal feedback [11]. Feedback was also measured directly by testing growth responses of different plant species to conditioned AM fungal communities **(f)**. Pairwise feedbacks, as represented by interaction coefficients [39], were calculated and used to test patterns of strength of feedback across host phylogeny and life history **(g)**. We also tested whether the observed changes in AM fungal species composition or changes in genetic composition within species predicted strength of mycorrhizal feedback **(h)**.

even within individual cells [43,44], we grouped amplicon sequence variants (ASVs) into species to permit assessment of changes in AM fungal species composition and changes in genetic composition of individual AM fungal species (Fig 1b). We tested for consistent effects across plant phylogeny and across plant life history categories with two approaches. Permutational Multivariate Analysis of Variance (PerMANOVA) controls for non-independence of amplicons in individual samples. Phylogenetic generalized mixed models (PGLMMs) explicitly test for host phylogeny influence on AM fungal composition and simultaneously control for phylogenetic non-independence. Both test for effects of host life history on AM fungal composition (Fig 1c).

We used two approaches to assess the ecological impact of changes in AM fungal composition (Fig 1). As our constructed mycobiomes included isolates that have been characterized in their impacts on host growth across plant family and across plant life history [23,35], we test for correlations of AM fungal fitness responses to hosts with AM fungal impacts on host growth (Fig 1d and 1e). A positive correlation of host and symbiont fitness impacts predicts positive, reinforcing feedback that would generate co-adaptation [11,45]. Secondly, we directly tested the functional consequences of host-specific differentiation in AM mycobiome composition using a feedback assay (Fig 1f) in which plant species were grown with the differentiated mycobiomes in full factorial combinations [39,46]. We then tested whether the strength of feedback is positively correlated with phylogenetic distance, as expected by co-adapted phylosymbiosis, and whether AM fungal feedback varies consistently with plant life history, as expected from mycorrhizal dynamics influencing species turnover during succession (Fig 1g). Finally, we tested whether changes in AM fungal species or genetic composition within AM fungal species generate resulting feedbacks (Fig 1h).

## Results

We found significant influence of plant phylogeny and plant life history on the species and genetic composition of the AM mycobiome. This holds true whether controlling for the multivariate nature of the amplicon data using PerMANOVA or whether testing for and controlling for phylogenetic relationships of host plants using PGLMMs. As these results are complementary, but redundant, we present the PGLMM results which provide a direct test of phylosymbiosis below and the PerMANOVA within the appendix (S2 and S3 Tables).

### Host phylogeny impacts on AM Mycobiome

Grouping ASV counts by individual AM fungal species we examined the effect of host relatedness on the overall AM fungal community. We found significant changes in AM fungal composition and density with plant phylogeny. Host phylogeny significantly affected AM fungal diversity and density in year 1 ($p < 0.05$, $p < 0.001$) and the relative proportion of *Cetraspora pellucida* in year 1 ($p < 0.05$), *R. fulgida* in year 2 ($p < 0.001$), and *Entrophospora infrequens* in both years ($p < 0.001$, $p < 0.05$, S4 Table and Fig 2). This suggests that phylogenetically proximate plant species have more similar AM fungal composition, reflecting more similar impacts of host on AM fungal relative fitness, than more phylogenetically distant plant species. *Ce. pellucida* had high relative growth rates with plants in Apocynaceae, *E. infrequens* had low relative fitness with grasses, and *R. fulgida* tended to perform best with grass species (Fig 2). We also analyzed phylogenetic heritability of AM fungal composition, i.e., the proportion of species differences in AM fungal composition attributable to host phylogeny. While highly variable across AM fungal species, the results confirm that AM fungal species composition and diversity were predicted by host plant relatedness (S5 Table).

### Host phylogeny impacts on genetic composition of AM fungal species

Using individual PGLMMs, we also tested the influence of host phylogeny on the genetic composition of individual AM fungal species. We identified 4 ASVs for *Cl. clariodeum*, 2 ASVs for *Claroideoglomus lamellosum*, 8 ASVs for *E. infrequens*, 15 ASVs for *A. spinosa*, 9 ASVs for *R. fulgida,* and *Funneliformis mosseae* has 12 ASVs. ASV's identified for each species were tested in order of decreasing overall abundance. We found significant differences in ASV composition due to plant

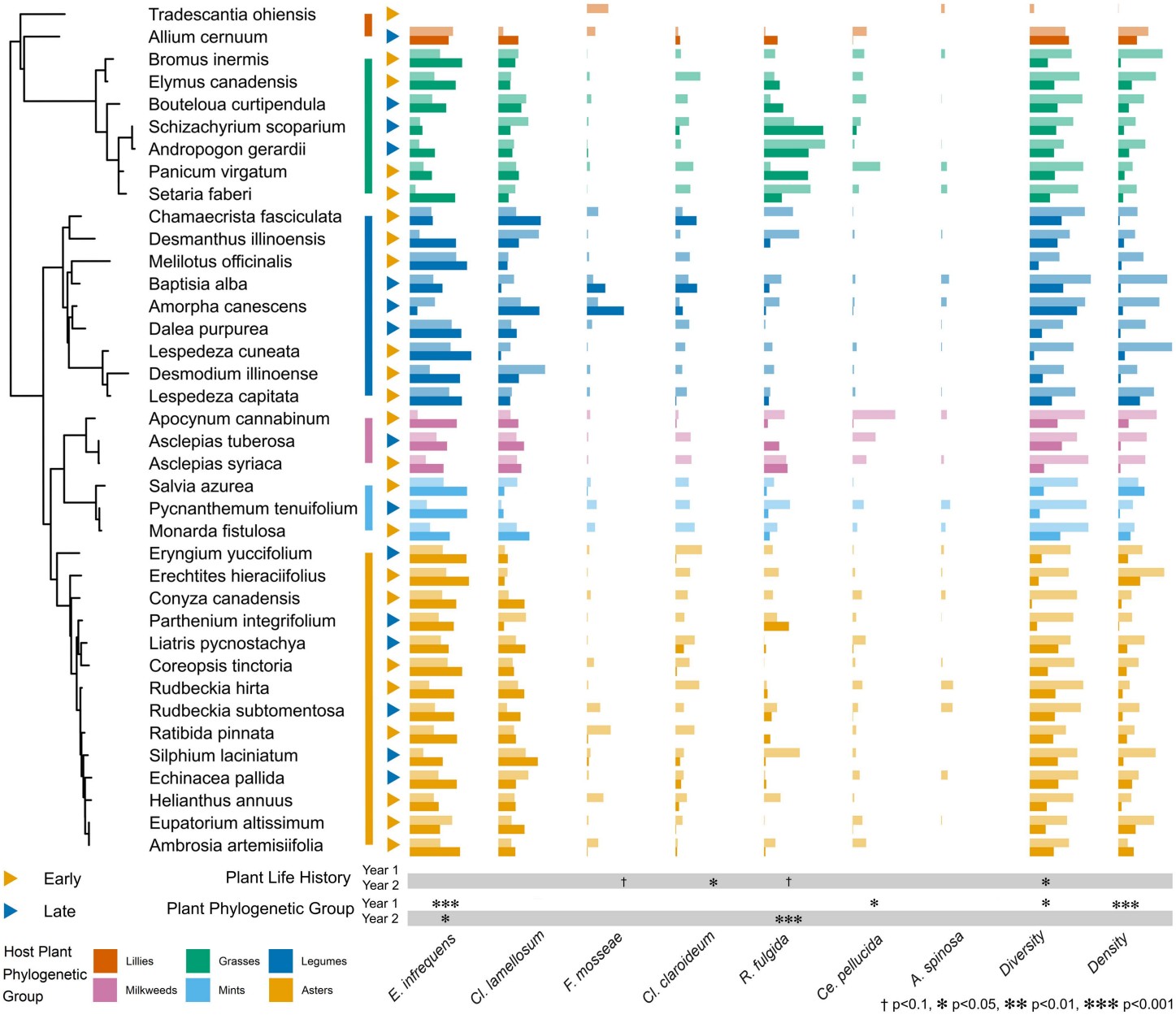

**Fig 2. AM fungal species relative abundances across plant phylogeny.** AM fungal species composition varied across plant phylogeny and life history. Relative abundances of *Claroideoglomus lamellosum, Claroideoglomus claroideum, Entrophospora infrequens, Funneliformis mosseae, Racocetra fulgida, Cetraspora pellucida, Acaulospora spinosa* are presented alongside estimated abundance (proportion of ASVs identified as inoculated over total AM fungal ASVs) and Shannon's Diversity. Host plant species are arranged by phylogeny with phylogenetic group indicated by vertical colored bars and life history status indicated by colored triangles adjacent to each plant species name. Significance of phylogenetic and life history effects on individual metrics of AM fungal composition are presented for each year separately. For each plant species, Year 1 and Year 2 means of AM fungal metrics are presented as grouped columns with year one being the lighter shaded top bar. The data and code underlying this Figure can be found in https://doi.org/10.17605/OSF.IO/NAXMT.

phylogenetic relatedness for the most abundant *E. infrequens* ASVs for both years ($p < 0.001$, $p < 0.05$, Fig 3). The most abundant ASV of *Cl. lamellosum* was predicted by plant phylogeny in year 1 with marginal statistical significance ($p < 0.1$). *F. mosseae's* and *Claroideoglomus claroideum's* third most abundant ASV as well as *Ce. pellucida's* fourth most abundant

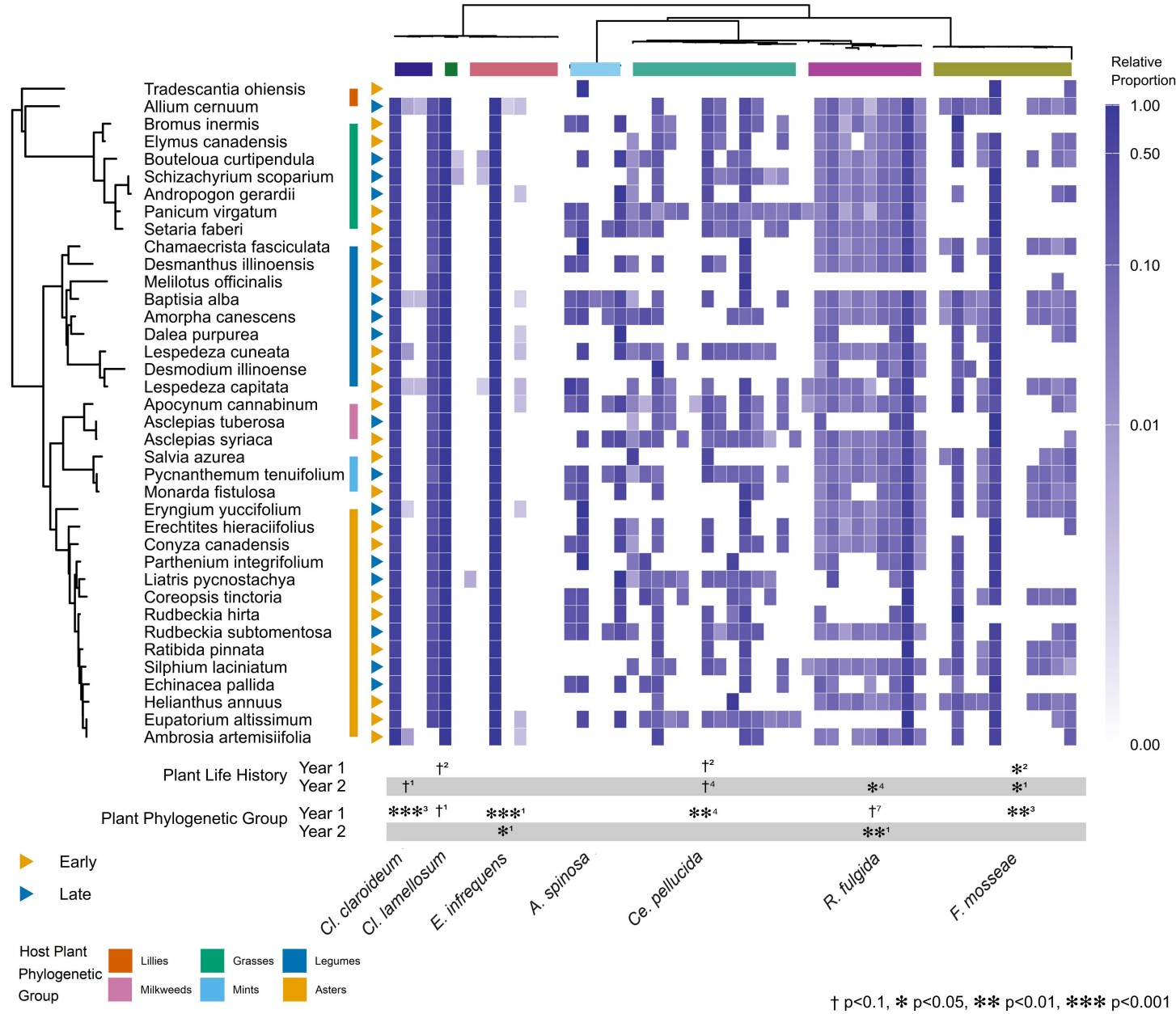

**Fig 3. AM fungal genetic composition across plant phylogeny.** AM fungal genetic composition (ASV relative abundances) varied with host phylogeny and life history. ASVs of *Claroideoglomus lamellosum*, *Claroideoglomus claroideum*, *Entrophospora infrequens*, *Funneliformis mosseae*, *Racocetra fulgida*, *Cetraspora pellucida*, *Acaulospora spinosa* are arranged according to the AM fungal phylogeny presented at the top of the figure, with ASVs belonging to different AM fungal species indicated by differently colored horizontal bars underneath the phylogeny. AM fungal species names are listed on the bottom of the figure. Host plant species are arranged as in Fig 2. For each host plant species, the proportion of ASVs for each individual AM fungal species is represented by color intensity on a log scale. The significance of phylogenetic and life history effects are presented for each year separately. For each AM fungal species, results are reported for the most abundant ASV to show a significant effect, with rank abundance of the ASV indicated by the superscript next to the significance symbols. The data and code underlying this Figure can be found in https://doi.org/10.17605/OSF.IO/NAXMT.

ASV were also predicted by host plant phylogeny in year 1 ($p < 0.01$, $p < 0.001$, $p < 0.01$, respectively). Finally, *R. fulgida* was predicted by phylogeny for its seventh most abundant ASV with marginal statistical significance in year 1 and greater confidence in year 2 ($p < 0.1$, $p < 0.01$, respectively). These results show that host plant relatedness had a measurable effect on AM fungal genetic composition for most AM fungal species.

## Plant life history impacts on AM mycobiome

PGLMMs of total ASV counts grouped by individual AM fungal species showed statistically significant changes in AM fungal composition with plant life history. The relative abundance of *F. mosseae*, *R. fulgida,* and *Cl. claroideum* were marginally or significantly higher when associated with late successional plant species in year 2 ($p < 0.1$, $p < 0.1$, $p < 0.05$, respectively, S4 Table and Fig 2). The four other AM fungal species did not show consistent differences with plant life history (S4 Table). Late successional plant species had more diverse AM fungal communities than early successional plant species (Shannon index, year 2 $p \leq 0.05$, S2 Fig). Estimated AM fungal density was not significantly affected by host plant life history in either year ($p > 0.1$).

## Plant life history impacts on genetic composition of AM fungal species

Host plant life history significantly impacted the genetic composition of several AM fungal species. We conducted a PGLMM analysis examining how plant life history category influenced genetic composition of individual AM fungal species as measured by changes in relative abundance. The analysis found the relative abundance of the second most abundant ASV of *Cl. lamellosum*, *F. mosseae*, and *Ce. pellucida* were each affected by host plant life history in year 1 ($p < 0.1$, $p < 0.05$, $p < 0.1$, respectively, Fig 3). In year 2, the first and fourth most abundant ASVs of *F. mosseae*, and the fourth most abundant *Ce. pellucida* ASV were significantly affected by host plant life history (all $p < 0.05$, Fig 3). Finally, the most abundant *Cl. claroideum* ASV had a marginal response to host plant life history in year 2 ($p < 0.1$). These results confirm that host plant life history significantly impacted genetic composition of multiple AM fungal species.

## Correlation of host and symbiont relative fitness

Previous work with the AM fungal isolates used in this experiment [23,35] allowed us to test for general patterns across four family groups: Asters (Asteraceae), Legumes (Fabaceae), Lillies (Amaryllidaceae and Commelinaceae), and Grasses (Poaceae). Meta-analyses identifies that AM fungal species differentially impact growth of plant species in different families (Wald Test QM = 69.3310, df = 27, $p < 0.0001$). We found that *E. infrequens*, *Cl. lamellosum*, and *Cl. claroideum* confer greater benefits to plants in the Asters relative to plants in the Grasses group (S3B Fig). We also saw that *E. infrequens*, *Cl. lamellosum*, *Cl. claroideum*, *F. mosseae*, *A. spinosa,* and *Ce. pellucida* confer greater benefits to plants in the Lilly group compared to the Grasses. These effects are weakly positively correlated with the relative fitness responses to these plant families (Fig 4). This provided weak evidence of mycorrhizal positive feedback particularly between plants in the comparison of the phylogenetic groups Asters and Grasses ($R^2_{yr1}$=0.32, $p = 0.187$, $R^2_{yr2}$=0.48, $p = 0.127$) as compared to Legumes – Grasses ($R^2_{yr1}$=0.32, $p = 0.188$, $R^2_{yr2}$=0.49, $p = 0.183$) and Lillies – Grasses ($R^2_{yr1}$=0.03, $p = 0.702$, $R^2_{yr2}$=0.01, $p = 0.877$). Overall, these meta-analysis results provide limited support for a strengthening of host fitness with phylosymbiosis. While none of these correlations were statistically significant, the power of these tests are limited by comparison of only seven AM fungal species.

We also found that growth promotion by AM fungal isolates differed between early versus late successional plant species (S3A Fig, Wald Test QM = 163.3303, df = 34, $p < 0.0001$). Moreover, there was a negative correlation between AM fungal growth response and benefits to plant hosts, with late successional plants growing less well with the AM fungal species that we found accumulated on late successional hosts ($R^2_{yr1}$=0.23, $p = 0.274$, $R^2_{yr2}$=0.94, $p = 0.001$, Fig 5). Notably, *E. infrequens* and *Cl. lamellosum*, the best growth promoters for late successional plant species, had the lowest growth rates with late successional plant species in our experiment. This pattern is consistent with changes in AM fungal composition generating negative feedback between

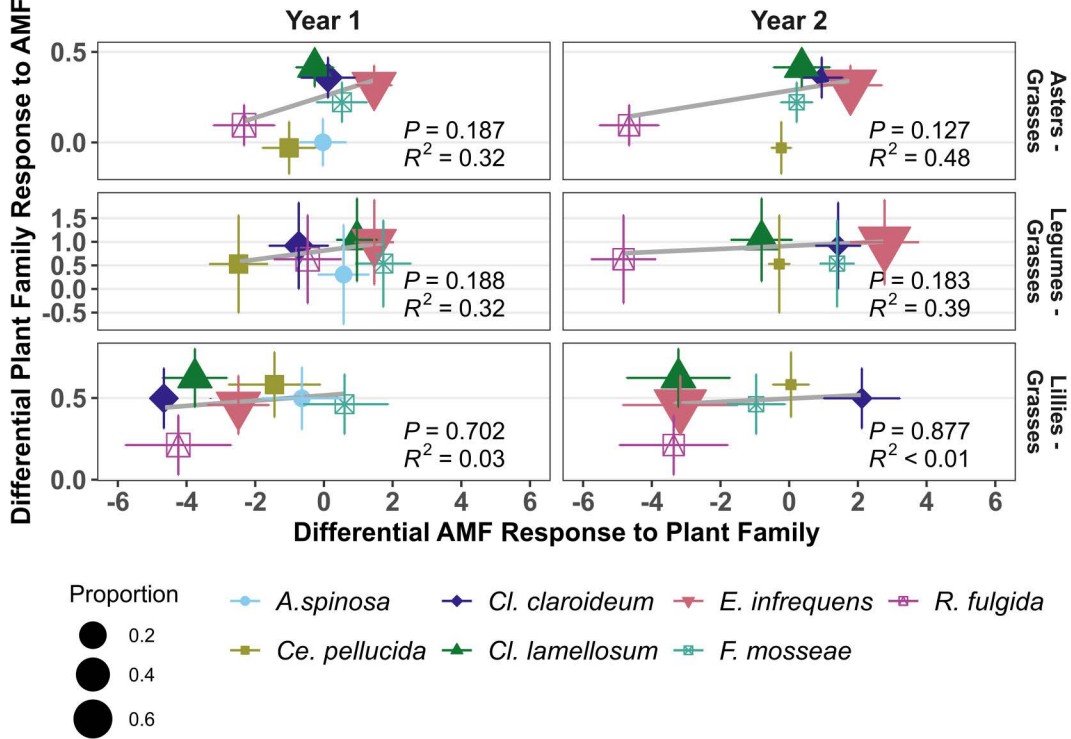

**Fig 4. Predicted feedbacks between phylogenetic groups.** We tested the correlation of differential growth response of plants of different phylogenetic groups to AM fungal species (derived from meta-analysis of previous growth assays), with the measured differential AM fungal accumulation in host plant phylogenetic groups (Fig 2). Positive slopes of the regression lines in gray suggest a strengthening of host fitness with phylosymbiosis, though there were no significant correlations at the $p \leq 0.1$ level. Panels are arranged by year at the top of the figure. Differential responses are calculated with plant response relativized to the grass family. $R^2$ and $p$ values corresponded to linear correlations weighted by AM fungal relative abundance. AM fungal species are denoted by color and symbol type. Symbol size denotes the relative proportion of each AM fungal species. The data and code underlying this Figure can be found in https://doi.org/10.17605/OSF.IO/NAXMT.

plants of different life history strategies. Given that AM fungal species diversity increased with late successional plant species in the second year of AM mycobiome training (Fig 2), it is possible that the change in AM fungal diversity feedback on fitness of late successional plant species. However, when analyzing two studies comparing early and late successional plant species in response to single and mixed inocula, we did not find a difference in response to AM fungal diversity between plant life history stages ($p = 0.748$, S4 Fig), suggesting feedback through changes in mycorrhizal diversity was unlikely.

## AM fungal feedbacks: Tests of patterns

Direct tests of AM fungal feedback between pairs of plant species varied from significantly positive to significantly negative (Fig 6). Across all pairs of plants, median ($p = 0.003$) and mean ($t = 3.9472$, $p \leq 0.001$) AM fungal feedback was generally positive (S5 Fig). However, there was no relationship between strength of feedback and plant phylogenetic distance. The best fit was a marginal quadratic fit ($x$: $p = 0.167$, $x^2$: $p = 0.085$, Fig 6A), as closely related species and most distantly related species tended to have zero or negative feedbacks. Together, this indicates that the observed plant phylogenetic influence on AM mycobiome composition did not consistently result in improved host fitness.

When analyzing feedback effects between plant families (restricted to the three best represented families: grasses, asters, and legumes). We tended to find mean positive feedback between legume species ($t_{13} = 2.233$, $p = 0.058$), negative feedback between Aster–Grass pairings ($t_{18} = -2.236$, $p = 0.029$) and Grass–Grass pairings ($t_6 = -2.082$, $p = 0.082$). Other

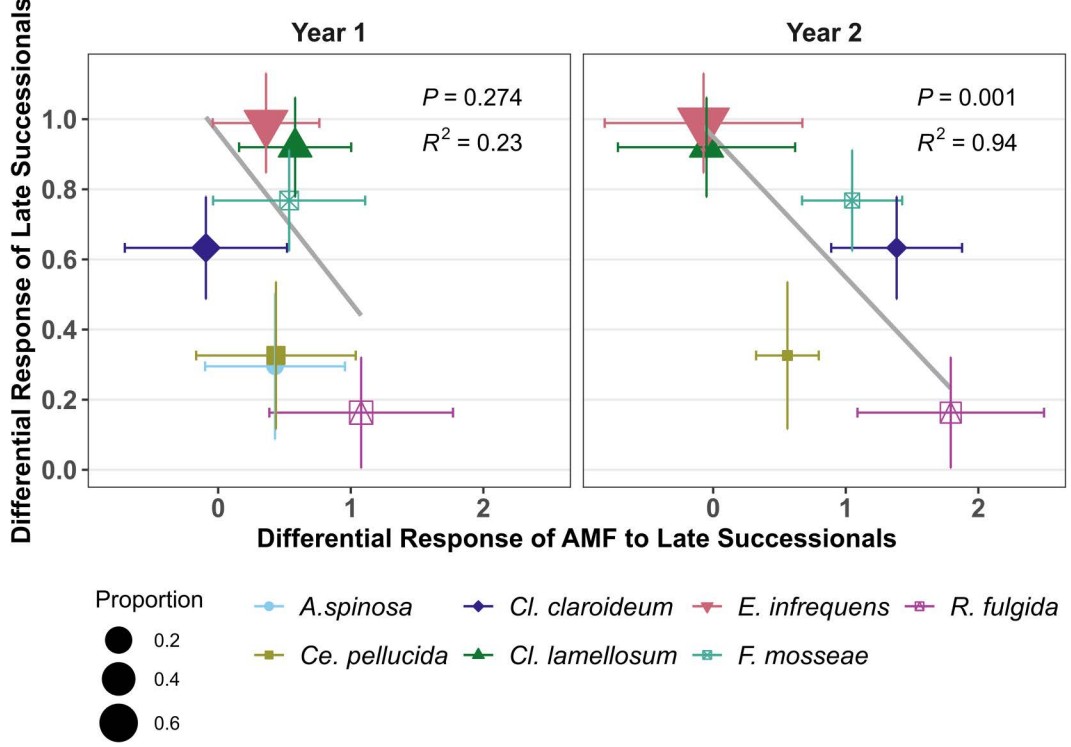

**Fig 5. Predicted feedbacks between life history categories.** The differential growth response of plants of different life histories to AM fungal species (derived from meta-analysis of previous growth assays) was negatively correlated with the differential AM fungal response to host plant life history category (Fig 2) in year 2 ($p = 0.001$), but not year 1. The negative correlation is consistent with changes in AM fungal composition generating negative feedback between life history categories. Panels are arranged by year at the top of the figure. Differential responses are calculated with plant response to early successional plant-associated fungal communities as the baseline. $R^2$ and $p$ values correspond to linear correlations weighted by AM fungal relative abundance. Gray lines are the regression lines. The data and code underlying this Figure can be found in [https://doi.org/10.17605/OSF.IO/NAXMT](https://doi.org/10.17605/OSF.IO/NAXMT).

comparisons were not consistently different from zero (S6 Fig). Median Legume–Legume pairings remained marginally positive ($p = 0.078$), and Aster-Grass remained significantly negative ($p = 0.014$). Grass–Grass pairings did not have significant negative median feedbacks ($p = 0.109$).

In comparing within and between plant life history categories, the median pairwise feedbacks between pairs of early successional species tended to be positive ($p = 0.069$), and the mean feedback was significantly positive ($t_{41} = 2.654$, $p = 0.01$, Fig 6B). Comparing feedback between early and late successional plant species, both the median ($p = 0.009$) and mean values ($t_{68} = 3.055$, $p = 0.003$) were significantly positive. When both plant species were late successional neither the median ($p = 0.488$) nor the mean ($t_{14} = -0.009$, $p = 0.993$) feedback were significantly different from zero.

### AM fungal feedbacks: Community and/or genetic drivers

Differences in AM fungal species composition and/or differences in genetic composition within AM fungal species could drive mycorrhizal feedbacks. Changes in AM fungal species composition, as represented by dissimilarity between plant species pairs, failed to predict feedback strength (Fig 7a, $R^2 = 0$, $F = 0.2922$, $p = 0.59$). However, changes in genetic composition of individual AM fungal species did predict mycorrhizal feedback. Specifically, positive pairwise feedback was significantly predicted by dissimilarity in genetic composition of the two most common AM fungal species: *E. infrequens* (Fig 7b, $R^2 = 0.13$, $F = 17.78$, $p < 0.001$) and *Cl. lamellosum* (Fig 7c, $R^2 = 0.1$, $F = 13.27$, $p < 0.001$). Genetic dissimilarity of other AM fungal species did not predict pairwise feedback (S7 Fig).

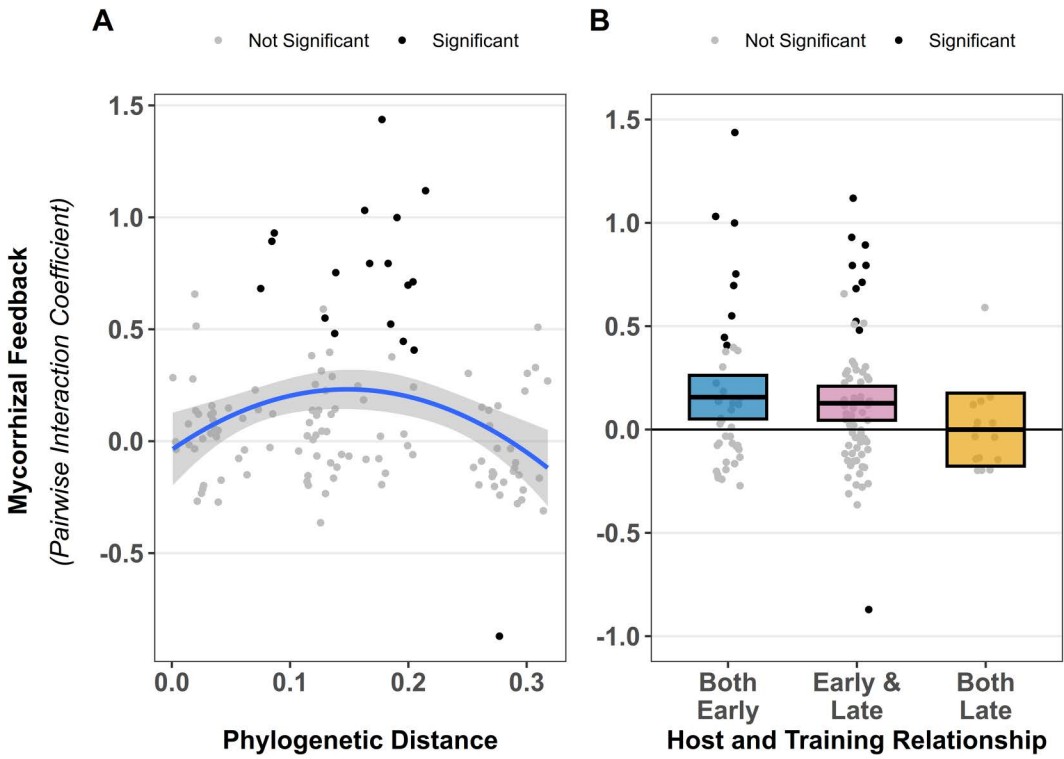

**Fig 6. Measured feedback strength between AM fungi and plant hosts.** Mycorrhizal feedback, measured as pairwise interaction coefficients [39], are plotted on the Y-axis in all plots. Phylogenetic distance **(A)** had a marginally significant quadratic relationship with fitness ($p = 0.085$). **(B)** shows feedbacks broken out by pairs of early, pairs of late, and early vs. late host plant parings. Bars indicate mean feedback, which were significantly positive for early–early ($p = 0.01$) and early-late ($p = 0.003$) parings. Both panels show individual interaction coefficients plotted as shaded dots, with gray for insignificant and black for significant pairwise interactions ($p > 0.05$). The data and code underlying this Figure can be found in https://doi.org/10.17605/OSF.IO/NAXMT.

## Discussion

Using 38 plant species, we present a uniquely robust test of plant phylogeny and life history impacts on the composition and symbiotic function of AM fungi. We found strong phylosymbiosis, as phylogenetic distance was an important predictor of the differences in AM fungal diversity, density, and community composition across plant host species. In addition, we detected significant differences in genetic composition of individual AM fungal species across plant host species indicating that AM fungal species can evolve rapidly in response to their host. These genetic differences were often predicted by plant phylogeny. However, while AM fungi generally improved host growth, we found only weak evidence that the phylogenetically structured divergence of symbiont community and genetic composition consistently altered symbiont impacts on their hosts. After controlling for the effect of plant phylogeny, we also found that AM fungal density, community diversity, community composition, and genetic composition within species changed with plant life history, as represented by early versus late successional stage categories. In this case, the shift in symbiont composition with plant life history differentially benefited plants of the same life history generating positive mycorrhizal feedback. Differential positive feedbacks that vary between plants of different life history characteristics could contribute to plant species turnover during succession. Interestingly, our results suggest that the genetic change within AM fungal species, rather than changes in relative abundance of AM fungal species, generated the observed positive mycorrhizal feedback. Together, our results identify that both plant phylogeny and plant life history traits influence the associated AM fungal community and genetic composition, but only changes induced by plant life history traits consistently generated reinforcing positive feedbacks on plant fitness.

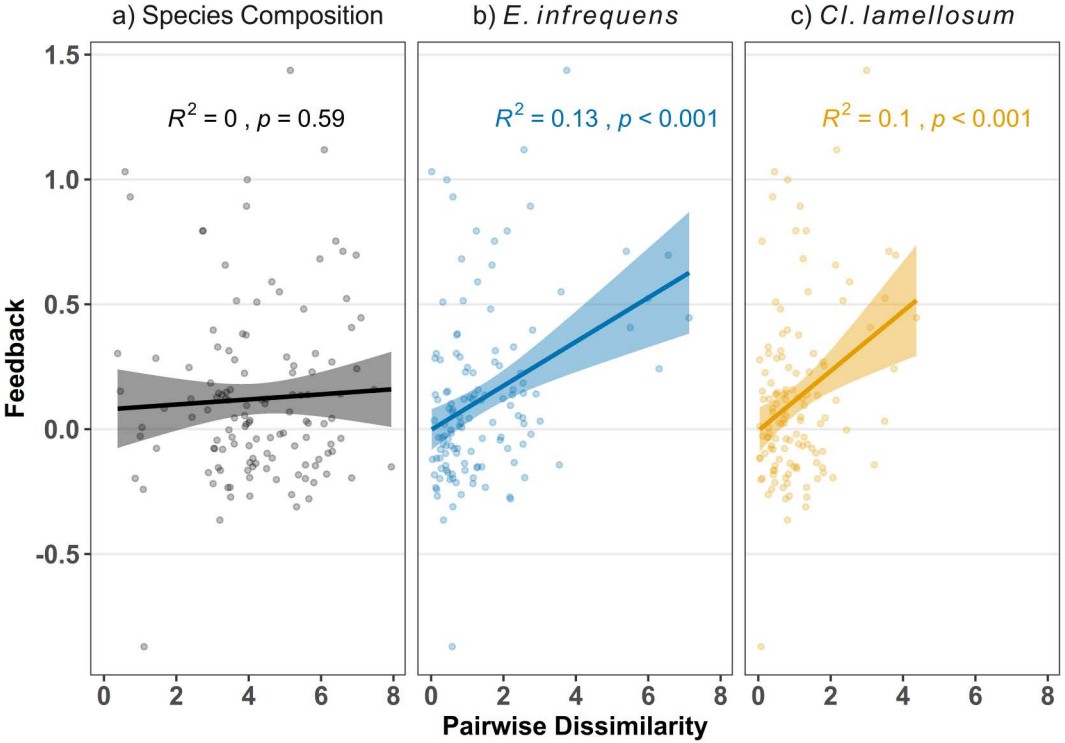

**Fig 7. Mycorrhizal feedback strength predicted by genetic dissimilarity of AM fungi.** Regressions of strength of pairwise feedback against measures of AM fungal dissimilarity. AM fungal species composition, as measured by pairwise dissimilarity of total species counts (Aitchison distance) did not predict feedbacks ($p = 0.59$). Pairwise feedback was significantly predicted by genetic dissimilarity of two AM fungal species, *Entrophospora infrequens* ($p < 0.001$) and *Claroideoglomus lamellosum* ($p < 0.001$). The data and code underlying this Figure can be found in https://doi.org/10.17605/OSF.IO/NAXMT.

Previous studies have demonstrated that AM fungal growth rates varied between individual plant species [46,47]. We affirm that this result is general across plant species of the tallgrass prairie system, as AM fungal composition diverged significantly across the 38 plant species and 8 families studied. Moreover, host-specific differentiation of AM fungal composition had a strong phylogenetic signature. That is, the growth rates of individual AM fungal species tended to be similar on phylogenetically similar host plants. This phylogenetic influence was evident in the differential abundance of *E. infrequens, R. fulgida*, and *Ce. pellucida* (Fig 2). Phylogenetic distance also predicted measures of AM fungal diversity. While previous work has shown phylogenetic signature to plant species-specific differentiation of soil bacterial and fungal communities [48], this study provides clear evidence of phylogenetic structure to host-specific differentiation of AM fungal communities.

We note that the influence of host identity on AM fungal composition does not appear to result from differential exclusion of AM fungal species from roots, as would be expected from incompatibilities of plant–AM fungal signaling. Rather, inoculated AM fungi were found with all plant species. Given that inoculated AM fungi had equivalent densities across each plant species at the beginning of the experiment, the change in AM fungal composition is due to host-specific differences in AM fungal growth rates (i.e., fungal fitness). The observation of substantial influence of host phylogeny on AM fungal composition may reflect that plant influence on AM fungal relative fitness can be moderated by phylogenetically conserved plant traits, such as root architecture [49] or secondary chemical composition [50].

Our first measure of host phylogenetic influence on mycobiome composition solely examined species-level changes. Host phylogeny could also influence genetic changes within each species. AM fungal species and isolates have highly variable rDNA genes [43,51], and we observed changes in rDNA composition with plant host species across most AM

fungal species included in this study. Using both PerMANOVA and PGLMM methods, we detected unique host family/phylogenetic structure of the genetic composition of all our AM fungal species except for *A. spinosa*, a species that had very low relative abundances. Our most abundant AM fungal species, *E. infrequens,* exhibited host phylogenetic influence on its genetic composition for both years, while *Cl. lamellosum*, *Cl. claroideum*, *Ce. pellucida*, *R. fulgida*, and *F. mosseae* exhibited phylogenetic responses in genetic composition in one year regardless of the analysis used. In addition, host plant life history showed at least a marginal effect on AM fungal genetic composition for all species except *E. infrequens* and *A. spinosa*. Previous work had shown that genetic composition of AM fungal species can change rapidly with environment and host plant species [28,52,53]. These rapid changes could be due to differential segregation of nuclei in multinucleate cells [27,28,33]. We provide evidence that such genetic change can occur widely across AM fungal taxa and that the host-specific effects on AM fungal genetic composition are structured by plant phylogeny.

We find host-specific changes in AM fungi composition generate feedback on plant growth that is, on average, positive and ranges from significantly negative to strongly positive for individual plant species pairs. This is consistent with previous studies [15,17,18]. Our study provides the first attempt to generalize the patterns of AM fungal feedback, specifically testing whether AM fungal feedback is structured by plant phylogeny or plant life history. We test these patterns by testing correlations of AM fungal differentiation with growth impacts of these fungi observed in previous growth assays with our AM fungal isolates [23,35]. In addition to this correlational approach, we also measured AM fungal feedback effects by analyzing patterns observed among feedbacks directly measured using an independent greenhouse assay [46]. While the correlation of plant and AM fungal fitness responses only considers the impacts of changes in AM fungal species composition, the direct measurements of AM fungal feedbacks integrates across changes in AM fungal species and genetic composition.

Both correlations of relative plant and fungal fitness (Fig 4) and patterns of AM fungal feedbacks (Fig 6C) show weak responses to plant phylogenetic structure. That is, the AM fungi with the highest growth rates on a particular group of phylogenetically-related plants do not generally promote the growth of these phylogenetically-related plants better than other AM fungi. Interestingly, the best fit of AM fungal feedbacks to plant phylogenetic relatedness was curvilinear, with regions of phylogenetic distance over which plants tended to benefit more from AM fungal communities cultivated by more distantly related host plants and regions where plants tended to benefit more from AM fungal communities cultivated by more phylogenetically similar host plants (Fig 6C). Given that this is the first test of such relationships, more work is necessary in the plant–AM fungal systems as well as other host–symbiont interactions.

In contrast to plant phylogeny, AM fungal feedbacks are structured by plant life history. AM fungal feedback, as measured by direct greenhouse assays, were consistently positive between early and late successional plant species (Fig 6B). This result is consistent with observations greater responsiveness of late successional species to mycorrhizal fungi [23,35], and of positive mycorrhizal feedback observed in mesocosms [18] and field inoculations with prairie AM fungi [26,54,55]. Positive mycorrhizal feedback could play an important role in plant dynamics during succession and restoration, potentially accelerating turnover during succession, or contribute to alternative stable states [11]. Mycorrhizal feedback between pairs of late successional plant species were not generally different from zero, consistent with all late successional species benefiting equivalently from changes in mycorrhizal composition, while feedback between pairs of early successional species tended to be positive, suggesting that this life-history category includes plants with substantial variation in interactions with AM fungi.

The change in AM fungal species composition was not sufficient to explain the positive AM fungal feedback between early and late successional plant species (Fig 5). In fact, across all the data, the change in AM fungal species composition did not predict observed patterns of mycorrhizal feedback (Fig 7a). AM fungal species diversity also increased with late successional plant species, and AM fungal diversity can influence host response [56] potentially generating positive AM fungal feedbacks. However, this hypothesis was not supported by meta-analysis of differential benefits to late successional plant species from AM fungal diversity (S4 Fig).

The observed patterns of mycorrhizal feedback appear to have resulted from changes in genetic composition within individual AM fungal species. This hypothesis is directly supported as the genetic change within the two most common and beneficial AM fungal species, *E. infrequens* and *Cl. lamellosum*, significantly predicted the strength of positive mycorrhizal feedback (Fig 7b and 7c). Our results are consistent with previous observation of host-driven genetic changes within AM fungal isolates altering AM fungal impacts on plant fitness [27,33], and of patterns consistent with a role of genetic variation within AM fungal species generating mycorrhizal feedback [16]. Our work illustrates that rapid genetic change within AM fungal species can have functional consequences, even exceeding those of changes in AM fungal species composition. Our result highlights limitations to efforts to understand AM fungal ecology using patterns of species level morphological traits [57,58] as we find that mycorrhizal feedback, the product of variation in AM fungal growth responses to hosts and impacts on hosts, meaningfully varied between genotypes within AM fungal species rather than between AM fungal species.

We find both ecological (i.e., changes in relative abundance of AM fungal species) and evolutionary (i.e., changes in genetic composition within AM fungal species) dynamics operating concurrently within the plant–AM fungal symbiosis. We find strong evidence that individual AM fungal species, and genotypes within species, vary in their response to plant species, and that patterns of AM fungal divergence were structured by both plant phylogeny and plant traits governing life history variation. Changes in AM fungal composition generally improved host fitness, i.e., mycorrhizal feedbacks were generally positive, consistent with coadaptation of plant and fungus. However, there was substantial variation in strength of mycorrhizal feedbacks, with variation in plant life history predicting stronger positive feedbacks, while variation due to plant phylogeny did not. Moreover, evolutionary changes within individual AM fungal species were the major driver of positive feedback rather than ecological changes in AM fungal species composition. Given the evidence that AM fungal diversity and composition can be critical to plant community and terrestrial ecosystem function [5–9], our work suggests that improved understanding of both evolutionary and ecological dynamics of AM fungi are necessary for optimal management of terrestrial ecosystems and projections of ecosystem response to anthropogenic disturbance.

## Materials and methods

A common community of AM fungi made up of seven cultured isolates derived from tallgrass prairie (*Cl. lamellosum*, *Cl. claroideum*, *E. infrequens*, *F. mosseae*, *Racocetra fulgida*, *Ce. pellucida*, *A. spinosa*) was distributed into replicate mesocosms. These isolates were inoculated from mixtures of pure inocula into two gallon pots of sterilized 50/50 sand soil mix. The mesocosms were planted in the greenhouse with one of 38 grassland plant species from the prairie region chosen to provide a high degree of both phylogenetic diversity as well as diversity in life history traits associated with plant successional stage (S1 Fig and S1 Table). AM fungal composition in mesocosms was allowed to differentiate in response to host species over two growing seasons (2017 and 2018) and the composition of the AM fungal community was sampled after each growing season. In both years plants were grown in the greenhouse for the entire growing season (March to September) and were harvested in the fall. Plants were grown in 152 two-gallon pots with four individuals of a species in individual pots in the first year. After the first year, half of the soil in each pot was retained as inocula for the same host species and same replicate in year 2. Initial soil training was conducted in a climate-controlled greenhouse. Temperatures ranged from 70–90 F, a broad-spectrum light system was used 14 hours a day. The greenhouse had a closed HVAC system and humidity was kept at approximately 50%. Pots had drip irrigation with approximately 70 ml of water administered daily.

Soil samples were taken after each growing season for the determination of AM fungal composition using MiSeq Illumina sequencing. We homogenized roots and soil from the entire pot and then sub-sampled 0.2 g for sequences. We assumed over-representation of sequences as indicative of positive fitness impacts in plant treatments and under-representation indicative of negative fitness impacts. In year 3, we conducted a greenhouse feedback experiment where each soil community, conditioned by association with specific plant species, was used as inocula for the 38 plant species

in the experiment. These were grown with a single host plant individual in 1-l pots, with 900 ml of 50/50 sterilized sand soil mix, and inoculated with 100 ml of soil from the year 2 mesocosms. From the plant biomass data, we were able to calculate feedback using interaction coefficients (see statistical methods for equation) [39]. The feedback experiment was conducted in a greenhouse cooled with a swamp cooler, exposed to ambient lighting conditions, and with drip irrigation.

## Meta-analysis methods

The seven AM fungi isolates used in this experiment were used previously in three separate studies testing their impact on early and late successional plant species: one reported in [35] and two in [23]. We conducted a meta-analysis of these studies to obtain estimates of the growth promotion for each isolate to plant species of different plant phylogenetic groups (grass, asters, and legumes) and from different life histories/origin groups (early successional native, non-native, and late successional native plants). We calculated the log mycorrhizal responsiveness (LRR) using equation (2) where $x_{inoc}$ and $x_{ctrl}$ are mean plant biomass for the inoculated treatments and sterile controls, respectively. The sampling variance ($\hat{\sigma}_2$) of LRR was estimated using equation (3) where $SD^2_{inoc}$ and $SD^2_{ctrl}$ are the standard deviations and $n_{inoc}$ and $n_{ctrl}$ are the sample sizes of the treatment and controls, respectively [59,60].

$$LRR = \ln\left[\frac{x_{inoc}}{x_{ctrl}}\right] \tag{1}$$

$$\hat{\sigma}_2 = \frac{SD^2_{inoc}}{n_{inoc} \times \overline{x}^2_{inoc}} + \frac{SD^2_{ctrl}}{n_{ctrl} \times \overline{x}^2_{ctrl}} \tag{2}$$

The analysis was conducted using the metafor (version 4.8) package in R (version 4.4.3) [61]. Moderators were the host plant life history, host plant phylogenetic group, the AM fungal isolate used, and the interaction of the isolate used and plant life history and phylogenetic group. Random effects included individual experiments and plant host species. The model was estimated using the restricted maximum-likelihood ("REML") method. Statistical significance was determined using the anova.rma() method for metafor which conducts a Wald Test of all predictors. Wald Tests were conducted on the full model as well as reduced models containing either plant life history or plant phylogenetic group. Marginal means were estimated for the interaction terms using the emmeans (version 1.11.2) package [62]. The metafor model was converted to a reference grid for compatibility with the emmeans package using the qdrg function.

We also tested whether plant life history predicted plant response to differences in AM fungal diversity of inocula. Only the two studies in Cheeke and colleagues [23] included multi-species with single AM fungal species inoculations. Following the approach developed by Magnoli and Bever [56], we calculated LRR in this analysis as the ratio of multiple species inocula mean plant biomass $x_{multi}$ to the mean plant biomass averaged across all single inocula $x_{mean}$. We also calculated the ratio of $x_{multi}$ to the mean plant biomass of the single most beneficial AM fungal inocula $x_{most}$. For each of these tests, we constructed measures of variance by modifying equation (2).

## Molecular methods

To measure the changes in relative abundances of AM fungal species 0.2 g of soil was retained from each pot for Illumina sequencing. DNA extraction was performed on 152 samples using the Qiagen DNeasy Powersoil Kit. One sample of the 152 was lost during storage in both years. DNA was amplified using the LROR and FLR2 primers [63,64]. Volumes of 12.5 μL of Phusion Hot Start Flex DNA Polymerase master mix, 0.5 μL of each primer, and 10.5 μL of MilliporeSigma Direct-Q 3 type I water per sample were combined with 1 μL of extracted DNA. PCR was conducted using one 5 min cycle at 94.0 °C, 35 cycles alternating between 94.0 °C for 30 s, 48.0 °C for 30 s, 72.0 °C for 30 s, and 72.0 °C for 10 min. Illumina adapters were attached using a second PCR run. Sequencing was performed on Illumina MiSeq 2x300.

## Bioinformatics methods

Bioinformatics was conducted using the Dada2 and Qiime 2 pipelines as described in [40–42]. Primers were removed using Cutadapt. Using Dada2 forward reads were truncated to a length where 95% of base pair Phred quality scores (Q scores) had a greater than 99% base call accuracy, with forward reads truncated to a length of 190 bp and reverse reads to a length of 140 bp. Forward and reverse reads were merged using the "justconcatonate" flag in Dada2 and chimeras were removed. Total sequencing depth ranged from 1,999 to 48,273 reads. Taxonomy was assigned by building a phylogenetic tree of the ASVs using RAxML together with an included library of known AM fungi species [40,41,43,65,66]. All AM fungi identified sequences were blasted against the reference library. Several ASVs that were placed inside AM fungi had unacceptably long branch lengths and were suspected of being nonhomologous gene regions that were erroneously being placed in the phylogeny [40]. Those ASVs were excluded from the analysis. All background soil was sterilized. Therefore, the ASVs that were not identified as one of the seven inoculated species were assumed likely to be legacy DNA (from the heat sterilized sand-soil mix) and were also not included in the statistical analysis. ASVs were assigned a species taxonomy if they clustered within known species sequences from the reference library. We used the proportion of inoculated AM fungal ASVs to overall sequencing depth as a proxy for fungal density. Unfortunately, a complete dataset of spore counts was not completed for all 152 training pots. The relationship between our proxy and spore count for those pots where data was available was statistically significant using a Poisson generalized linear model ($p < 0.001$, S8 Fig). The low McFadden pseudo $R^2$ (0.051); however, shows that this is a weak proxy. We include it in the final analysis as a best estimate and note that our overall conclusions remain consistent with our without the consideration of fungal density.

## Plant phylogenetic methods

A phylogeny was constructed for all plants in the study to determine phylogenetic distance. Sequences were obtained for the rbcL gene for all 38 species of plants from the NCBI GenBank database. A constraint tree was obtained using the Phylomatic "Tree of Trees" software (version 3) using the slik megatree [67,68]. Tree construction was conducted using RaXML (Version 8) [66].

## Statistical methods

ASV abundances for inoculated AM fungal species were centered log transformed and a PerMANOVA analysis was conducted using the adonis function in the R package vegan (version 2.7) [69]. The predictors used were host plant life history, host plant phylogenetic group, and the interaction of host plant life history with host plant phylogenetic group. Phylogenetic groups are characterized as lilies (Amaryllidaceae and Commelinaceae), grasses (Poaceae), legumes (Fabacea), milkweeds (Apocynaceae), mints (Lamiaceae and Plantaginaceae), and asters (Apiaceae and Asteraceae).

Using a categorical factor, such as phylogenetic group, to analyze the effect of phylogenetic relationships is a limited approach. We used a method to include phylogenetic distances as a random effect to preserve the information included in the full phylogenetic tree. A PGLMM was fitted using the phyr (version 1.1) R package [70,71]. Host plant successional status, and the interaction between them were modeled as fixed effects. Greenhouse block and the log transformed AM fungal DNA sequences were included alongside phylogenetic distance and species as random effects. As a univariate analysis each AM fungal species' ASVs were grouped, and their abundances logit transformed. We also assessed Shannon's Diversity and the logit-transformed proportion of the inoculated AM fungi over the total number of ASVs. Shannon's Diversity was calculated on rarefied AM fungal sequence read counts (sample size of 100). We use the proportion of inoculated AM fungi as a proxy for changes in AM fungal density, as it correlates with total spore counts from a subsample of mesocosms (S8 Fig), consistent with the background DNA not changing in response to plant hosts.

We purposely included full factorial tests of plant species and training plant species for subsets of the 38 plant species. Each subset was treated as a block, and we attempted to maximize comparisons across the phylogeny, while maintaining

a relatively even number of both early and late successional plants (S9 Fig). Within these sub-set pairs of plant species, we estimated pairwise feedbacks that govern the influence of mycorrhizal fungal dynamics on plant–plant interactions. Pairwise feedbacks are measured by the interaction coefficient ($I$), as derived from the models of interguild frequency dependence and host-microbiome feedback. It is calculated according to equation (3) [11,39]. Where $F_{A\alpha}$ is the fitness of plant $A$ in soil $\alpha$, $F_{A\beta}$ the fitness of Plant $A$ in soil $\beta$, $F_{B\alpha}$ the fitness of plant $B$ in soil $\alpha$, and $F_{B\beta}$ the fitness of plant $B$ in soil $\beta$. In this experiment we used plant biomass as our measure of plant fitness. If this coefficient is positive then the plants are experiencing positive feedback, and if the coefficient is negative the plants are experiencing negative feedback. We used a type III ANOVA analysis to test the effect on the interaction coefficient from host plant and training plant life history and the interaction between them.

$$I = F_{A\alpha} - F_{A\beta} - F_{B\alpha} + F_{B\beta} \tag{3}$$

Dissimilarity between species compositions and genetic compositions within each AM fungal species was calculated using Aitchison distances (Euclidean distances based on center log-transformed counts). For species composition, all ASVs assigned to a single AM fungal species were first summed. For genetic change dissimilarity, separate dissimilarity matrices were calculated for all ASVs characterized as belonging to each AM fungal species. These dissimilarities were used in the PerMANOVA analyses. The list of pairwise dissimilarities derived from these matrices were used in regression analysis against feedback strength (interaction coefficients)

We also tested the influence of plant phylogeny on AMF composition as phylogenetic heritability (sensu [72]), represented using the ratio of variance explained by the phylogeny over total variation between plant species, using equation (4). In the analysis, $\hat{\sigma}_a^2$ was equal to the variance estimate for the phylogenetic distance matrix and $\hat{\sigma}_T^2$ was equal to the sum of the variance explained by phylogenetic distance and species effects.

$$H_P^2 = \frac{\hat{\sigma}_a^2}{\hat{\sigma}_T^2} \tag{4}$$

## Supporting information

**S1 Fig. Host plant characteristics.** Plant species were selected to represent a phylogenetically diverse group of early and late successional plants. Plants were selected from 5 large phylogenetic groups; grasses (Poaceae), legumes (Fabaceae), asters (Asteraceae and Apiaceae), mints (Lamiaceae and Plantaginaceae), milkweeds (Apocynaceae and Asclepiadaceae), and lilies (Liliaceae and Commelinaceae).
(DOCX)

**S2 Fig. Marginal means of relative abundance by plant life history.** These are the estimated marginal means of AM fungal relative abundances when host plants were early or late successional. AM fungal species are arranged top to bottom in order of most to least beneficial. The data and code underlying this Figure can be found in https://doi.org/10.17605/OSF.IO/NAXMT.
(DOCX)

**S3 Fig. Meta-analysis results.** The marginal means for the log response ratio's (LRT) of each of the seven AM fungal species used in this study. The LRT is the natural log of the ratio between how large plant species grew with AM fungal inoculation versus how they grew in a sterile control. These are derived from a meta-analysis of three previous experiments using these same cultures. **(A)** shows late successional plant benefits when grown in association with *E. infrequens*, *Cl. lamellosum*, *F. mosseae*, and *Cl. claroideum*. **(B)** shows the marginal means for the log response ratio's (LRT)

of each of the seven AM fungal species used in this study by host plant phylogenetic group. Asteraceae experienced significantly greater growth than Poaceae in association with all Claroideoglomus species (including *E. infrequens*). We also saw significant differential greater growth in Liliaceae over Poaceae as well as Liliaceae over Asteraceae in several AM fungal species. The data and code underlying this Figure can be found in https://doi.org/10.17605/OSF.IO/NAXMT. (DOCX)

**S4 Fig. LRR by life history of AM fungal mixtures to single symbiont.** These are the log response ratios of the mean plant biomass for plants inoculated with a mixture of AM fungi to the mean plant biomass representative of the average single symbiont and to the most effective single symbiont respectively. Each panel breaks our results depending on if the host plant was an early or late successional. The data and code underlying this Figure can be found in https://doi.org/10.17605/OSF.IO/NAXMT. (DOCX)

**S5 Fig. Histogram of mycorrhizal feedback measured as pairwise interaction coefficients.** This shows counts of all paired comparisons with both the median and mean being significantly positive ($p = 0.003$, $p \leq 0.001$). The data and code underlying this Figure can be found in https://doi.org/10.17605/OSF.IO/NAXMT. (DOCX)

**S6 Fig. Feedbacks by plant phylogenetic group.** Mycorrhizal feedback, as measured by the pairwise interaction coefficient, compared across phylogenetic group pairings. Feedback between individual pairs of plant species are indicated with dots, with pairwise feedback that was significantly different from zero indicated in black. We note that pairwise mycorrhizal feedbacks ranged from significantly negative to significantly positive. On average, we observed significantly positive mycorrhizal feedback between two legume species ($p \leq 0.01$) and significant negative mycorrhizal feedback between aster and grass species ($p \leq 0.05$). The data and code underlying this Figure can be found in https://doi.org/10.17605/OSF.IO/NAXMT. (DOCX)

**S7 Fig. Mycorrhizal feedback strength predicted by genetic dissimilarity of AM fungi for all AM fungal Species.** Regressions of strength of pairwise feedback against measures of AM fungal genetic dissimilarity. Pairwise feedback was significantly predicted by genetic dissimilarity of two AM fungal species, *E. infrequens* ($p < 0.001$) and *Cl. lamellosum* ($p < 0.001$). The data and code underlying this Figure can be found in https://doi.org/10.17605/OSF.IO/NAXMT. (DOCX)

**S8 Fig. Correlation between proxy AM fungal density and spore count.** The proxy for AM fungal density is the proportion of inoculated AM fungal ASVs and sequencing depth. This was used because spore counts were not completed for all 156 training pots. The relationship between our proxy and spore count for those pots where data was available was statistically significant ($p < 0.001$). The data and code underlying this Figure can be found in https://doi.org/10.17605/OSF.IO/NAXMT. (DOCX)

**S9 Fig. Feedback experimental design.** The experimental design for the feedback experiment included 5 subset groups of plants where all plant pairings were made in a fully factorial design. Each group also included early and late successional plants. When these pairings are arranged phylogenetically it becomes clearer that we also have a good representation of species pairs across the plant phylogeny. This allows us to test pairwise feedbacks, host plant characteristics, and plant phylogeny effects in a single experiment. The data and code underlying this Figure can be found in https://doi.org/10.17605/OSF.IO/NAXMT. (DOCX)

**S1 Table. Plant species used.**
(DOCX)

**S2 Table. PerMANOVA results for AM fungi species.** To test for overall differences in AM fungal composition with plant life history and plant family, we conducted a PerMANOVA on centered log ratio (CLR) transformed counts of ASVs grouped by AM fungal species. This analysis reveals statistically significant effects on AM fungal species composition of plant phylogenetic group in both years ($p \leq 0.001$, $p \leq 0.001$). We also tested for overall differences in AM fungal composition with plant life history and plant family using the PerMANOVA approach. This analysis reveals statistically significant effects on AM fungal species composition of plant life history on AM fungal species composition in year two ($p < 0.001$).
(DOCX)

**S3 Table. PerMANOVA results for AMF ASVs variation within species.** As AM fungi are known to have substantial variability in rDNA composition within individual multinucleate cells, we also test for changes in ASV composition within individual AM fungal species using a PerMANOVA approach. For *E. infrequens* and *Cl. claroideum*, we find evidence of change in genetic composition with plant phylogenetic group in both year one ($p \leq 0.001$, $p \leq 0.01$) and year two ($p \leq 0.1$, 0.01). Genetic composition of *Cl. lamellosum* and *Ce. pellucida* showed significant responses of plant phylogenetic group in year 1 ($p \leq 0.001$, $p \leq 0.001$), while genetic composition of *F. mosseae* and *R. Fulgida* had significant effects in year 2 ($p \leq 0.01$, $p \leq 0.001$). The genetic composition of *F. mosseae*, *Cl. claroidium*, and *R. fulgida* showed an effect of host plant life history on intraspecies ASV variation in year 2 ($p < 0.001$, $p < 0.01$, $p < 0.05$).
(DOCX)

**S4 Table. PGLMM results for AM fungi species.** Phylogenetic generalized linear mixed model (PGLMM) results for relative proportions of each AM fungal species (combined ASV counts), as well as Shannon Diversity and Logit transformed density estimates.
(DOCX)

**S5 Table. Phylogenetic heritability.** Phylogenetic heritability of microbiome composition describes the differential influence of plant species on microbiome composition as predicted by the plant phylogeny. A substantial proportion of variation, ranging from 0.000% for *Cetraspora pellucida* to 99.999% for *Entrophosphora infrequens*, in species impacts on microbiome composition that can be explained by plant phylogeny.
(DOCX)

## Acknowledgments

We are grateful for the assistance of Sheena Parsons, Alice Tipton, Liz Koziol, Terra Lubin, Alexa Phillips, Reagan Smith, Kristen Mecke, Kelly Chesus, and the many undergrad technicians for assistance in project initiation, inocula preparation, and the planting, harvesting, and processing of the feedback experiment.

## Author contributions

**Conceptualization:** Robert J. Ramos, Peggy A. Schultz, James D. Bever.

**Data curation:** Robert J. Ramos, James D. Bever.

**Formal analysis:** Robert J. Ramos, James D. Bever.

**Funding acquisition:** Peggy A. Schultz, James D. Bever.

**Investigation:** Robert J. Ramos, James D. Bever.

**Methodology:** Robert J. Ramos, James D. Bever.

**Project administration:** Robert J. Ramos, James D. Bever.

**Visualization:** Robert J. Ramos, Brianna L. Richards.

**Writing – original draft:** Robert J. Ramos, James D. Bever.

**Writing – review & editing:** Robert J. Ramos, Brianna L. Richards, Peggy A. Schultz, James D. Bever.

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
