## [Editor Report · Decision Letter 0]

3 Jul 2025

Dear Dr Ramos,

Thank you for submitting your manuscript entitled "Plant phylogeny and life history predict AMF species and genetic composition, but only life history and genetic composition predict feedback" for consideration as a Research Article by PLOS Biology.

Your manuscript has now been evaluated by the PLOS Biology editorial staff, as well as by an academic editor with relevant expertise, and I'm writing to let you know that we would like to send your submission out for external peer review.

Once your full submission is complete, your paper will undergo a series of checks in preparation for peer review. After your manuscript has passed the checks it will be sent out for review. To provide the metadata for your submission, please Login to Editorial Manager (https://www.editorialmanager.com/pbiology) within two working days, i.e. by Jul 07 2025 11:59PM.

Kind regards,

Roli Roberts

Roland Roberts, PhD

Senior Editor

PLOS Biology

rroberts@plos.org

---

## [Decision Letter · Decision Letter 1]

22 Sep 2025

Dear Dr Ramos,

Thank you for your patience while your manuscript "Plant phylogeny and life history predict AM fungal species and genetic composition, but only life history and genetic composition predict feedback" went through peer-review at PLOS Biology. Your manuscript has now been evaluated by the PLOS Biology editors, an Academic Editor with relevant expertise, and by three independent reviewers. Please accept my apologies for the extremely long time that the process has taken over the summer months.

You'll see that reviewer #1 says that the work is rigorous, but thinks that the manuscript needs to be clarified and streamlined. Reviewer #2 is also positive, but similarly found the manuscript hard to follow, and suggests some re-structuring; s/he also wants much more methodological detail, and some additional data. Reviewer #3 finds the work “compelling and well conducted” but also complains about the clarity, especially of the methods section.

In light of the reviews, which you will find at the end of this email, we are pleased to offer you the opportunity to address the comments from the reviewers in a revision that we anticipate should not take you very long. We will then assess your revised manuscript and your response to the reviewers' comments with our Academic Editor aiming to avoid further rounds of peer-review, although we might need to consult with the reviewers, depending on the nature of the revisions.

**IMPORTANT - SUBMITTING YOUR REVISION**

*Resubmission Checklist*

*Published Peer Review*

*PLOS Data Policy*

*Blot and Gel Data Policy*

Sincerely,

Roli Roberts

Roland Roberts, PhD

Senior Editor

PLOS Biology

rroberts@plos.org

REVIEWERS' COMMENTS:

Reviewer #1: PBIOLOGY-D-25-02002R1

Here the authors present a manuscript investigating plant host relatedness and AM fungal diversity and composition using an experimental setup of many grassland and fungal species. This research builds onto existing work on host-symbiont relations, and is novel in that the author's investigate mycorrhizal feedbacks onto host plants, over different years. Overall I find the research rigorous, however a few notes could improve the manuscript to fit well with the broad readership of PLOS. The main concern I have is digesting so many different methods into one clear story.

First, the manuscript can be considerably shortened, for example in the introduction covers many other symbioses than plant-AM fungal, but the results here are largely discussed in context of plants and AM fungi. It is a lot of work reading manuscripts so making it as clear and easy for the reader is a major goal of convincing them to continue onto the next paragraph of the paper. In this vain it would help to maybe have an introduction where it is structured in 1-2 paragraphs of the main problem and walk the reader through the different ways you test this, and why you test is in so many ways (as show in in Fig 1.)

Likewise the use of a meta-analysis is welcome because it builds on previous research but no information is given on the power/confidence in these results. Lastly, the use of complex phylogenetic models is an advancement, but they also require more careful testing of your hypotheses, of which I suggest an addition to explain variance by phylogeny vs species effects uncorrelated to the host phylogeny, see reference in Ecol Letters below.

Due to the high number of experiments and analyses, it would benefit the text to use as little jargon as possible (e.g. abbreviations, I count at least 7 excluding the species names), different statistical tests names, plant-fungal species names). For example I see growth rates and fitness used through the manuscript (is this LRR?), please clearly define all terms especially if they are used to make claims in the discussion.

I also suggest using active or passive voice consistently in the document.

Specific Comments:

46: I would define symbiont, since this is in review in a general journal

59: The jump here is quite quick from phylosymbioses exist to these phylosiybioses are important for fitness. When symbiosis could also be commensal and not influence fitness. A sentence on why the host may influence fitness and "composition", which I assume is community composition is needed. Guiding the reader through the logic will put less strain on the reader to understand why they should even care about testing for influences on fitness/competition. This sentence may actually do well to start the next paragraph.

104: I am very happy to see more complex phylogenetic models being employed here.

140: Why include the PERMANOVA at all, if you know the other model is better equipped to answer your question?

142: Is a benefit of your mixed effect model being overlooked? If I understand your model structure correctly you can compare variance explained by phylogeny vs species identity here by looking at intraclass correlation coefficients. The ability to compare variance explained by phylogenetic relatedness of hosts vs species effects uncorrelated to host phylogeny would be welcome. Following: https://onlinelibrary.wiley.com/doi/full/10.1111/ele.13263 .

144: What is density? Of hyphae, biomass, spores?

Figure 2: Define the color bar in the caption please.

165-166 "third most abundant ASV had a significant effect of host plant phylogenetic distance in year 1 (p<0.01, p<0.001, p<0.001)."

179: How does one measure marginally significant higher? Or do you have greater confidence in sig based on the p-value?

Maybe I missed it but here and in other areas breaking this sentence down what does it mean if an ASV has an effect on host phylo distance. I would consider if effect is the correct word choice, because it is not clear what an ASV effect on phlyo distance would be.

Discussion General: It is clear much work has been done in this manuscript, and to help the reader it would be good to refer to the figures with your main discussion points. There is quite a lot of information, tests, and analyses here so really walking the reader through and in the discussion pointing back to specific parts of the results is welcome. This occurs once for Figure 2, but it is helpful to do more throughout. This is especially helpful for discussion paragraph one where you recap all the work done and distil it for the reader.

Further, here it seems that mycorrhizal feedbacks may also be time dependent. If fungal and host phenology are aligned, then would you expect +/- correlations. Given that AM fungi grow from carbon that has to be captured by the plant host and then traded to the fungi, it would be expected that hyphae and spores grow lagged compared to high plant productivity. It would be worth discussing the effects of time in a brief 1-2 sentences, and if the time of your sampling may bias measurements of feedback. E.g. line 339.

211: Here and in other places remind the reader what you are doing at each test, since this is so many things in one manuscrtip. "Next we conducted a meta-analysis of XXX to answer YYY". Our results show ZZZ.

215: Is this worth reporting, if so explain why.

281: Uniquely robust I believe is true in that many tests have been completed, but see concerns here about meta-analysis power as well as concerns over methods used to test for phylogentic signals (and comparison to species null models).

287: To support this statement it would be helpful to actually test this using the method I suggest in like 142. To compare the variance explained by host phylogeny. Unless this is referring to the meta-analysis?

299: I suggest to say "suggest" since these models are not explaining all variance.

304: Grass plant species? Or those studied here? All plant species feels like a very large claim

309-310: See comment for 142

318-319: This is assuming host-microbe fitness is aligned, which may not always be the case. See: https://onlinelibrary.wiley.com/doi/10.1111/eva.13333

323: It is worth saying this is also conducted withing a single plant family, and it is unclear how the dynamics you describe here play out at diff scales, e.g. within species, between orders.

361: What type of system would be the next best test?

415: how many years of divergence?

423: Nex-gen is now here for over 10 years, I think we can just say what type of Illumina sequencing.

429: Define the equation please

431: With only three studies, what type of statistical power is there to provide confidence in results? Please report the power for the meta-analysis, and mention in the main text. I suspect it is quite low and results should be interpreted with caution.

461: See above, what type of Illumina sequencing to what depth and where was it conducted?

501: I suggest adding in the ICC as mentioned above for these models, to not just give p-values for the phylogenetic relationships but determine magnitude of effect.

505: Is Shannon density telling you much here? If this analysis does not add much to your results I suggest removing and simplifying your story.

506-509: This is a very long sentence that I think should be broken up and explained clearly if this is referring to calculation of growth rates. Please define density. In general I suggest splitting the stats part into subsections based on what you are testing with what data and what tests.

524: Maybe I am missing it, but what if the fitness measurement, or is it a proxy like biomass?

5298: Is this an established method to look at genetic dissimilarity with grouped species from ASVs? Does this speak more to our poor species concept for AM fungi?

Figure captions: Many people only look at figures and captions, can you please write to show the main findings for the figure, what data and tests were used. Great figures stand alone.

Reviewer #2:

Summary:

This study investigates the role of plant phylogeny and life history in structuring the symbiosis between arbuscular mycorrhizal fungi and their herbaceous host plants. In particular, the study takes an experimental glasshouse-based approach in which a common set of 7 AMF species are mixed into soils containing 1 of 38 plant species and they are grown for two seasons. Periodic sampling of the soils is done to estimate AMF relative abundances over the experiment period, which is assessed with high-throughput sequencing. Additionally, a meta-analysis is conducted to estimate the strength of feedbacks for the 7 AMF with some of the hosts used in the experiment based on previous research in this system. The primary research question is whether plant phylogeny, plant life history, or both explain AMF community metrics as well as the strength of plant-AMF feedbacks.

The authors contextualize their study in work in other study systems that have shown that phylogenetic relatedness is an important factor affecting the structure and functioning of symbioses, for which they use the term phylosymbiosis. Given the ecological ubiquity and importance of the AM symbiosis, it is surprising that this has not received more explicit attention, making the relevance of this work high despite the greenhouse setting.

The analyses are structured around testing how change in AMF density and composition is driven by either or both plant phylogeny and life history. This involves applying either phylogenetically explicit GLMs or PERMANOVAs, which give general similar results. The results are also structured at the ASV level, so they allow for assessment of within AMF species variation in performance, which is referred to AMF genotypes. Following this primary analysis, the result then shift to incorporating the results of the meta-analysis to test how the strength of the AMF-plant feedback is impacted by plant phylogeny and life-history. This is primarily done with regression-based analyses.

Given the large number of plants and AMF, it perhaps not too surprising that there is significant variation in the AMF community across the hosts in the different years. The authors find support for effects of both plant phylogeny and life history on AMF community dynamics, but they find limited support for the strength of AMF-plant feedbacks to be clearly structured by phylogeny. There is some significant signal for positive feedbacks with life history, particularly with late successional species, which is consistent with those plants and AMF potentially being more co-dependent. Overall, the results do support some level of importance for both plant phylogeny and life history on AMF community dynamics and on AMF-plant feedbacks, but that there are many genotype, species and clade level patterns that make this relationships complex.

Major comments:

Possibly due the manuscript structure for review, I found the study surprisingly hard to follow. On the plant side, given so much previous work done in this system and possible even with the plants in this experiment (it seems like this may not be the only publication from this experiment?), I wanted to have a clearer explanation in the introduction about the features of this system that make it such a strong test of the question the authors are addressing. Currently, the introduction treats topics at a very generic level targeting any biologist interested in these questions, but given the genotype, species, and clade level patterns observed, I think a more study system specific introduction would be helpful.

On the fungal side, I was also unsatisfied with both the description of the sampling and analyses of the molecular data. For example, how was the soil in the pots sampled, from what depths, and how big were the pots. 0.2 grams of soils is a very small amount, so I could imagine where in the pot the sample was taken from could matter a lot. Regarding the molecular analysis itself, I appreciate the ASV level resolution, although I would like to know the number of ASVs per species were present in the spores to start with. It is very different across the 7 AMF species? The relative abundance analyses are also problematic for comparisons across taxa given the compositional nature of high-throughput sequencing (HTS), so some accounting for compositionality seems required to be sure the results of increases in one taxon are not resulting in compensatory decreases in another taxon just because the total number of sequences is fixed. The authors talk about AMF densities - how is that defined? That should be done with qPCR or some direct measure or quantity, not HTS data. There is little discussion of the fact that spores could be present as sources of DNA by the inoculated species (either residual spores or newly produced spores) so relative abundances reported might be affected by multiple parts of the life cycle of plants. Finally, given all the previous work in this system by these authors, I also wanted to see some discussion of the relationship between soil abundance and root colonization, as both seem important to feedback strength. While these may seem like minor details only important to mycologists, I think without more detailed discussion of the biology of the system itself, it very hard to evaluate the validity of the larger conclusions drawn.

Minor comments:

Some redundancy in the writing - particularly in the discussion. Circling back to take home messages nearly every paragraph is distracting and I suggest the authors try to focus on clean intro and concluding summary paragraphs in the discussion, with the other paragraphs being more specifically targeted at contextualizing the results relative to other similar studies.

Figure 1 has panels labeled with AMF fitness and Plant Fitness. As far as I can tell, neither number of spores or number of seeds was ever measured in this study. As such, AMF performance and Plant performance seem like more appropriate names.

There are small grammatical issues throughout the manuscript that need careful proofing. Examples: infrequens misspelled in a few places, Figure 4 Y axis spelling of response.

Reviewer #3:

The study by Ramos et al. investigates the phylogenetic and plant life history effects on AM fungal species abundance and composition. It is very interesting that they also look at the effects on AM fungal species genetic composition, which is rarely addressed. Overall I find the work compelling and well conducted, but parts of writing needs revising, especially adding more details to Methods sections. I also highlight a few points regarding life history traits of the chosen AM fungal isolates, that might influence preferential hyphal biomass allocation between soil and root compartments, while only soil was sampled.

Line 146-148 This sounds more like Discussion and there are other sentences like that in the results.

Line 283 It's hard to judge strength of the model just based on p value. Is it possible to provide (marginal) R2-values for the PGLMM models?

Line 295-296 Check the sentence for clarity.

I find it curious that out of the 7 AMF taxa used in this study, 2 (R. fulgida and Ce. pellucida) are edaphophilic (making high biomass in soil), 3 are rhizophilic (making high biomass inside roots; C. lamellosum, C. claroideum, F. mosseae) and 2 are ancestral (with no particular preference; E. infrequence, A. spinosa) and both edaphophilic and one ancestral end up being the responsive ones. AMF life history traits are according to Weber et al. 2018. Since soil was sampled and not roots, I'm wondering why only soil was sequenced and not roots and how this affects the conclusions?

Overall the Materials and Methods section needs some elaboration and basic details.

Please add more details about how the inocula were made, can you be sure that these are single-species isolation, how were they propagated, I guess they were not sterile?

How were the mesocosms inoculated - how much inoculate (grams or spores?) per isolate per pot?

What was the potting soil?

How many pots were there in total?

Line 410-413 What was the selection criteria for choosing these particular AM isolates?

Line 420 use SI units instead of gallon

What is over-representation and under-representation of sequences? Compared to what?

Line 426 not sure the word "trained" captures the meaning, perhaps "conditioned" is better?

Line 427 what other plants?

Line 428 Liter does not constitute depth

Line 444 add functions and versions for all mentioned packages

Line 446 phylogenetic instead of phylogenic

Line 452-454 citation style not complying with journal standards

Bioinformatics Methods - first paragraph is about molecular methods, consider making a separate paragraph on Molecular Methods. Please revise for clarity- E.g line 461 "Extraction was performed…" Extraction of what? PCR conditions are described, but could you please specify how is this related to Illumina sequencing? These do not seem to contain tagged primers? What about Illumina chip that was used? 2x250 or 2x300bp?

Line 478- 481 What was the proportion of non-target sequences? It could also be assumed that legacy DNA might be rather degraded after sterilization (which method was used?), thus not really amplifying long reads. Could the non-targeted DNA also originate from isolates, thus not having single-species but sometimes mixed-species isolates due to contamination issues etc? Sequencing isolates themselves would verify this. Also, did you sequence plant roots - this would also tell you if the non-targeted DNA was from live AMF or not? If you had control pots with only sterile pots, it would be possible to track contamination from spore dispersal through air etc.

Were the PCR product concentrations equalized before sequencing? How would that influence read abundances?

Was the sequencing data standardized? How much did sequencing depth vary between samples?

---

## [Decision Letter · Decision Letter 2]

16 Jan 2026

Dear Dr Ramos,

Thank you for your patience while we considered your revised manuscript "Plant phylogeny and life history predict AM fungal species and genetic composition, but only life history and genetic composition predict feedback" for publication as a Research Article at PLOS Biology. This revised version of your manuscript has been evaluated by the PLOS Biology editors, the Academic Editor, and two of the original reviewers.

Based on the reviews, we are likely to accept this manuscript for publication, provided you satisfactorily address the remaining points raised by the reviewers and the following data and other policy-related requests.

IMPORTANT - please attend to the following:

a) Please provide a more comprehensible Title. We suggest: "Host plant phylogeny predicts the composition of arbuscular mycorrhizal fungal communities but not the mutualistic benefits" (the current title is too complex to be easily parsed, and includes punctuation and an abbreviation that is not widely known)

b) Please address the remaining requests from reviewer #1.

c) Please re-number the Supplmentary files sequentially within each type, i.e. your Supplementary Figs should be S1-S9, and Supplementary Tables S1-S5.

d) Please address my Data Policy requests below; specifically, we need you to supply the numerical values underlying Figs 2, 3, 4, 5, 6AB, 7ABC, S6, S8, S9, S10, S11, S12ABCDEF, S13, S14, either as a supplementary data file or as a permanent DOI’d deposition. I note that you already have an associated OSF deposition, but I was not able to access it using the link provided. Please provide this, and confirm whether the data and code in this deposition are sufficient to recreate the Figures?

e) Please cite the location of the data clearly in all relevant main and supplementary Figure legends, e.g. “The data underlying this Figure can be found in S1 Data” or “The data underlying this Figure can be found in https://zenodo.org/records/XXXXXXXX"

f) Please make any custom code available, either as a supplementary file or as part of your data deposition.

We expect to receive your revised manuscript within two weeks.

*Published Peer Review History*

*Press*

Sincerely,

Roli Roberts

Roland Roberts, PhD

Senior Editor

rroberts@plos.org

PLOS Biology

DATA POLICY:

Regardless of the method selected, please ensure that you provide the individual numerical values that underlie the summary data displayed in the following figure panels as they are essential for readers to assess your analysis and to reproduce it: Figs 2, 3, 4, 5, 6AB, 7ABC, S6, S8, S9, S10, S11, S12ABCDEF, S13, S14. NOTE: the numerical data provided should include all replicates AND the way in which the plotted mean and errors were derived (it should not present only the mean/average values).

CODE POLICY

Per journal policy, if you have generated any custom code during the course of this investigation, please make it available without restrictions. Please ensure that the code is sufficiently well documented and reusable, and that your Data Statement in the Editorial Manager submission system accurately describes where your code can be found. More information on our Code Policy, what and how to share can be found here: https://journals.plos.org/plosbiology/s/code-availability

DATA NOT SHOWN?

REVIEWERS' COMMENTS

Reviewer #1:

The authors have done considerable work to improve the readability of the manuscript and support the large amount of experimental work and analyses. I have a few minor comments.

AM fungal density:I am not fully persuaded that this is a reliable proxy. If this metric is commonly used, please add a citation. If not, stronger justification is needed. Related to this, I could not find a clear description of the model used for Fig. S13 in the Methods or in the figure caption. The fitted line in Fig. S13 has very narrow uncertainty, whereas the scatter of observed points looks very weak. If the relationship between spore counts and fungal density is weak, then confidence in density as a proxy should be low. Please revisit Fig. S13, provide the model details (name, variance explained, and clarify the definition and rationale of the density metric in the Methods (around line 535).

Figures:Several figure legends would benefit from more depth in explanation of the figures. For example, in Figure 5 it is not clear what the grey line represents. Please specify whether this is a fitted model, mean trend, or other summary and screen all figures for clarity. Please check spelling in Fig S9. In Supplemental be consistent with figure naming, some are title case others are not.

Fig S9. 99.999% seems like a very unique number, can this be confirmed? Is this a property of the model, since it is essentially 100% of variance explained it feels very high.

Methods:Please report greenhouse growing conditions, including temperature and light.

Please explain how model fits were assessed (e.g. man

Line ~533: "Hints" is not sufficient for a scientific manuscript, rewrite.

Lines ~550-565: Many transformations to data have been conducted in this section (log, logit) and is unclear if this was necessary. Such transformations of data should only be done if there is concern over violating model assumptions. Suggest saying assumptions were tested (e.g. residual normality) and then including transformations only if necessary.

Line 577: Is block the greenhouse block? Likewise "log transformed sequences" are the DNA sequences? Or many I am not following, please clarify.

Reviewer #3:

I feel that the authors have done a great job making the manuscript clearer, especially the methods section. I am satisfied with their responses to my comments (I was reviewer no 3) and have no further issues.

---

## [Editor Report · Decision Letter 3]

10 Feb 2026

Dear Dr Ramos,

Thank you for the submission of your revised Research Article "Host plant phylogeny predicts arbuscular mycorrhizal fungal communities but plant life history and fungal genetic change predict feedback" for publication in PLOS Biology. On behalf of my colleagues and the Academic Editor, Cara Haney, I'm pleased to say that we can in principle accept your manuscript for publication, provided you address any remaining formatting and reporting issues. These will be detailed in an email you should receive within 2-3 business days from our colleagues in the journal operations team; no action is required from you until then. Please note that we will not be able to formally accept your manuscript and schedule it for publication until you have completed any requested changes.

IMPORTANT:

a) Your new Title was too long, so we have partially reverted it to your previous Title, namely "Host plant phylogeny predicts arbuscular mycorrhizal fungal communities but plant life history and fungal genetic change predict feedback" - hopefully this is a reasonable compromise.

b) I've asked my colleagues to include the following request alongside their own: Please cite the location of the data clearly in all relevant main and supplementary Figure legends (i.e. those for Figs 2, 3, 4, 5, 6AB, 7ABC, S2, S3, S4, S5, S6, S7ABCDEF, S8, S9), e.g. using the sentence: “The data and code underlying this Figure can be found in https://doi.org/10.17605/OSF.IO/NAXMT" (yes, I'm afraid this might look repetitive, but it will make your Figures and their legends more standalone; make sure to use the OSF link, not the GitHub one)

Sincerely,

Roli Roberts

Senior Editor

PLOS Biology

rroberts@plos.org